# An Ultrasound-Guided Latero-Ventral Approach to Perform the Quadratus Lumborum Block in Dog Cadavers

**DOI:** 10.3390/ani13132214

**Published:** 2023-07-06

**Authors:** André Marchina-Gonçalves, Francisco G. Laredo, Francisco Gil, Marta Soler, Amalia Agut, José Ignacio Redondo, Eliseo Belda

**Affiliations:** 1Escuela Internacional de Doctorado, Programa en Ciencias Veterinarias, Universidad de Murcia, 30100 Murcia, Spain; 2Departamento de Medicina y Cirugía Animal, Facultad de Veterinaria, Universidad de Murcia, 30100 Murcia, Spain; laredo@um.es (F.G.L.); mtasoler@um.es (M.S.); amalia@um.es (A.A.); ebelda@um.es (E.B.); 3Departamento de Anatomía y Anatomía Patológica Comparada, Facultad de Veterinaria, Universidad de Murcia, 30100 Murcia, Spain; cano@um.es; 4Departamento Medicina y Cirugía Animal, Facultad de Veterinaria, Universidad Cardenal Herrera—CEU, CEU Universities, 46115 Valencia, Spain; nacho@uchceu.es

**Keywords:** abdominal analgesia, fascial block, latero-ventral approach, locoregional anesthesia, quadratus lumborum block, TAP block

## Abstract

**Simple Summary:**

The quadratus lumborum (QL) block is an ultrasound-guided regional anesthesia technique that aims to provide analgesia to the abdomen. Previous studies show that this technique may fail to provide analgesia in the cranial abdominal wall in dogs. The QL block is a difficult technique with the risk of damaging vital structures. The objective of this study is to assess a new latero-ventral approach to the QL (LVQL), which could be easier and safer to perform than the interfascial approach (IQL) widely reported. Another objective is to evaluate the distribution of high-volume injections (0.6 mL/kg) of dye/contrast achieved by both approaches. Six thawed dog cadavers were used to randomly perform the LVQL in one hemiabdomen and the IQL in the other. Distribution of the injectate throughout the ventral branches of the spinal nerves from T10 to L4 and the sympathetic trunk (CT and dissection) was assessed. The two approaches consistently stained L1-L3 ventral branches, which is compatible with obtaining somatic analgesia in the middle and caudal abdominal wall. The administration of high-volume injections did not result in a more cranial distribution of the injectate compared to previous studies. The sympathetic trunk was more consistently stained by the IQL, therefore, the LVQL may not promote visceral desensitization of the abdomen.

**Abstract:**

The QL block is a high-level locoregional anesthesia technique, which aims to provide analgesia to the abdomen. Several approaches of the QL block have been studied to find out which one allows a greater distribution of the injectate. The aim of this study is to compare the traditional interfascial QL block (IQL) with a new latero-ventral approach (LVQL). We hypothesize that this new approach could be safer and easier to perform, since the injectate is administered more superficially and further away from vital structures. Our second objective is to assess whether a higher volume of injectate (0.6 mL/kg) could reach the ventral branches of the last thoracic nerves, leading to a blockade of the cranial abdomen. Six thawed canine cadavers (12 hemiabdomens) were used for this purpose. Both approaches were performed in all cadavers. A combination of methylene blue/iopromide was administered to each hemiabdomen, randomly assigned to the LVQL or IQL. No differences were found regarding the ease of perform the LVQL with respect to IQL. The results show that both techniques reached the ventral branches from L1 to L3, although only the IQL consistently stained the sympathetic trunk (5/6 IQL vs. 1/6 LVQL). The use of a higher volume did not enhance a more cranial distribution of the injectate.

## 1. Introduction

The quadratus lumborum (QL) block is an ultrasound-guided regional anesthesia technique that aims to block the ventral branches of the thoracolumbar nerves and sympathetic trunk [1]. Several studies have been published in the last few years describing the QL block in dog cadavers and its potential analgesic effects [2,3,4,5,6]. These studies assessed different approaches to perform the QL block (Figure 1), and different volumes of injectate aiming to increase the desensitized area and provide visceral analgesia. Results from these studies indicate that the QL block is a feasible technique in dogs, able to provide somatic and visceral analgesia of the medium and caudal abdomen. However, none of these studies were able to consistently stain the ventral branches of the last thoracic spinal nerves (T10–T13) responsible for the innervation of the cranial abdominal wall. This fact indicates that the QL block would not be able to induce adequate desensitization for surgeries carried out in the cranial abdomen. A clinical study showed that dogs which received a QL block, as part of an opioid free anesthetic protocol for ovariohysterectomy, did not need analgesic rescue during the procedure nor within the next 4 h after surgery [7]. In the same report, thermal threshold testing demonstrated desensitization of the T10, T13 and L3 dermatomes 30 min after the surgery. These findings are controversial when compared with results from canine cadaveric studies that showed a more reduced distribution of the abdominal wall analgesia [2,3,4,5,6].

The QL block was first described by Blanco et al. [8] as a variation of the transverse abdominal plane (TAP) block. The TAP block has been widely studied in veterinary medicine in the last decade, and different approaches were described. Schroeder et al. [9] tested a mid-abdominal technique, which can promote an effective blockade of the ventral branches from T12 to L3. Similar results were obtained by Bruggink et al. [10]. In order to expand the distribution of the injectate, several approaches to the TAP block have been tested with one [11,12], two or three points of injection [13,14,15]. A major difference between QL and TAP blocks is that the QL block could reach the sympathetic trunk, promoting visceral analgesia [16], while the TAP block only provides somatic analgesia [17].

Quadratus lumborum block is a complex technique with a significant risk of damaging abdominal structures [1,18]. The sites of injection of the QL block are close to the abdominal aorta, caudal vena cava, spinal nerves, sympathetic trunk, and abdominal organs. Furthermore, due to the depth of the injection site, the visibility of these structures may not be adequate, particularly in larger or obese dogs. For these reasons, it is recommended that this technique is only performed by clinicians experienced in ultrasound-guided blocking techniques.

This study intends to assess a new approach to the QL block, in which the injectate is administered lateral and ventral to the quadratus lumborum muscle (LVQL), closer to the skin surface and further away from important abdominal structures, to increase the ease and safety of the technique. A second objective is to assess the distribution of a high volume of injectate (0.6 mL/kg), comparing the LVQL to the interfascial QL block (IQL) as described by Garbin et al. [2]. We hypothesized that the LVQL would be an easier technique to perform than the QL block, and that the higher volume of injectate administered would produce a more cranial distribution of the dye/contrast solution compatible with analgesia of the cranial abdomen in both approaches.

## 2. Materials and Methods

The Biosafety Committee in Experimentation (CBE 433/2021) and the Ethical Committee for Animal Experimentation (CEEA 740/2021) of the University of Murcia approved this essay. Six canine cadavers (12 hemiabdomens) were used in this research. Cadavers with skin or muscle lesion in the lateral aspect of the thoracolumbar area, or vertebral alterations were excluded from the study. The animals were humanely euthanized for reasons unrelated to the study and immediately frozen. The cadavers were thawed at room temperature 48 to 72 h before the procedures.

Two different approaches to the QL block (LVQL and IQL) were compared. Both approaches were performed in all the dogs. The technique to be performed in each hemiabdomen was randomly assigned, which was determined by extracting a code from a sealed envelope. To perform the study, cadavers were placed in lateral recumbency and an extensive clipping of the thoracolumbar area was carried out. The injections were made under ultrasound guidance, using a linear array (3–13 mHz, MyLab Gamma, Esaote, Florence, Italy) and echogenic needles (Ultraplex 10 mm 30°, BBraun, Melsungen, Germany). A volume of 0.6 mL kg^−1^ of a solution made up of the same amount of methylene blue (5 mg mL^−1^ Panreac Quimica, AppliChem, Castellar del Vallès, Spain) and radiopaque contrast medium iopromide (300 mg mL^−1^, Ultravist300, Bayern, Berlin, Germany) was injected in each hemiabdomen. The same researcher (Eliseo Belda) performed all the injections.

### 2.1. Ultrasound-Guided Technique

To perform both approaches, the ultrasound probe was placed transversal to the longitudinal axis of the spine, caudal and parallel to the last rib, at the level of L1. The array was slipped dorsally and tilted down until visualization of the target structures: erector spinae muscles (ESP), external oblique muscle (EO), internal oblique muscle (IO), transverse abdominal muscle (TA), transverse process of L1 (TP), transverse abdominal muscle aponeurosis (TAp), QL muscle, psoas minor muscle (PM) and vertebral body of (VB) L1, could be achieved (Figure 2).

Once those structures were identified, the needle was inserted in plane in a ventro-dorsal, latero-medial direction. After passing over the EO and IO, the tip of the needle was directed towards the TAP which offered slightly more resistance. The perforation of the TAP generates a “pop” sensation. Sonographic images and videos were recorded, and their quality was subjectively analyzed later (André Marchina-Gonçalves) as good (all the anatomical structures were sonographically identified), moderate (QL and PM margins were not clearly identified) and poor (anatomical structures beyond the TAP could not be visualized). The visualization of the needle tip at the moment of the injection was also registered.

#### 2.1.1. LVQL

The targeted injection site in this approach was the latero-ventral aspect of the QL, on its border with the TAP. To perform the LVQL, the needle was stopped immediately after piercing the TAP, and the dye/contrast solution injected (Figure 2).

#### 2.1.2. IQL

The injection site of this approach was the interfascial plane situated between the QL and PM [2]. This plane was observed as a hyperechoic line, formed by the junction of the fascias of both muscles. After drilling the TAP, the needle was guided further dorso-medially, passing through the QL. An increase in the resistance was noted again when the tip of the needle pierced the QL fascia. After a new “pop” sensation, the interfascial plane was reached and the solution injected (Figure 2).

### 2.2. Tomographic Study

Following the administration of the injectate, cadavers were taken to a computed tomography (CT) scan (High-Speed Dual, General Electric health care, Madrid, Spain) to evaluate contrast distribution. To perform the CT, cadavers were placed in sternal recumbence and 3 mm thickness slices were obtained at vertebral levels from T8 to L7. Images were subsequently reconstructed and analyzed by two radiologists (Marta Soler and Amalia Agut).

### 2.3. Dissection Study

Finally, the cadavers were subjected to anatomical dissection (Francisco Gil and André Marchina-Gonçalves), the technique is explained in detail elsewhere [6]. Briefly, the cadavers were placed in a dorsal recumbency and the ventral line was opened along the thoracolumbar area. After the opening of the abdominal cavity and sternotomy, abdominal and thoracic organs were evaluated for the presence of dye. At this point all the viscera were removed, and the structures of the retroperitoneal space identified. Then, a careful dissection was performed to analyze the distribution of dye in the retroperitoneal space, sympathetic trunk, ventral branches of the spinal nerves from T10 to L4 and genitofemoral and femoral nerves. Nerves were considered to be adequately blocked when all of their 4 quadrants (entire circumference) were stained to at least 1 cm in length.

### 2.4. Statistical Analysis

The statistical tests were performed using SPSS, version 24.0 (SPSS Inc., Chicago, IL, USA). A Shapiro–Wilk test was used to assess the normality of the distribution. The distribution of dye/contrast in the CT and anatomical dissection studies between the two treatments were evaluated by a Yuen’s test. The quality of the ultrasound images and visualization of the needle tip were analyzed by a Fisher’s test. Mean ± standard deviation, median (range) or numbers of animals were chosen to express the data as deemed most appropriate in each case. Data are expressed as mean ± standard deviation, median and range or number of animals, as it was considered more appropriate in each case. Differences were considered statistically significant when *p* < 0.05.

## 3. Results

Cadavers weighed 27.7 ± 12.5 kg and presented a median body condition score (BCS) of 4 (3–5) out of 9.

### 3.1. Ultrasound-Guided Injection

#### 3.1.1. Images Quality

The LVQL technique resulted in images classified as moderate (1/6) or good (5/6) accordingly to their quality. The IQL approach produced images of moderate (3/6) or good (3/6) quality. There were no statistical differences in image quality between the studied approaches (*p* = 0.5). The target structures were found in all cases.

#### 3.1.2. Needle Visualization

The tip of the needle was observed at the time of injection in 6/6 cases for LVQL and in 5/6 for IQL. No statistically significant differences were found between the two approaches (*p* = 0.85). In the case where the needle tip could not be observed, the correct positioning of the needle was confirmed after feeling two “pop” sensations as described before, as well as for the expected distribution of the injectate between QL and PM during the injection. 

### 3.2. Tomographic Study

Contrast was visualized on the target areas in all the hemiabdomens. In the LVQL, the contrast distribution extended 6.5 (4–8) VB, from T13 to L7, and in the IQL, 7.5 (5–9) VB, from T12 to L7 (Figure 3). No statistically significant differences were found between approaches (*p* = 0.74). In the LVQL, the contrast media was mainly distributed within the ventral and lateral aspects of the QL. Regarding the IQL, the contrast media was observed surrounding the QL and PM muscles, as well as the sympathetic trunk area (Figure 4). One hemiabdomen in the LVQL approach showed a wide distribution between the OI and TA muscles. Contrast media frequently reached the retroperitoneal space enhancing peri-renal tissues with both approaches. No contrast media was observed, neither in the intra-abdominal or intra-thoracic cavities, or in the epidural space.

### 3.3. Dissection Study

The LVQL approach resulted in a lateral distribution of dye, mainly staining the latero-ventral aspect of the QL, near the TA muscle. During dissection of the hemiabdomens subjected to the IQL approach, a distribution of dye around all the aspects of the QL and PM was observed (Figure 5). Distribution of dye in the retroperitoneal space was frequently observed in both approaches. In no case was methylene blue present in the abdominal and thoracic organs or cavities. In the LVQL, 3 (0–4) ventral branches, from L1 to L4 were stained vs. 4 (3–5), from T13 to L4 in the IQL (Figure 3). Statistical differences were observed only in relation to the staining of the L4 nerve (*p* = 0,03). The ventral branches of the spinal nerves in the LVQL were stained in a more distal portion, while the IQL dyed the nerves closer to their roots (Figure 5). The genitofemoral nerve was stained in 3/6 of the IQL vs. 1/6 in LVQL. The femoral nerve was found dyed in 1/6 IQL group and 0/6 in LVQL. Finally, the sympathetic trunk was stained in 0 (0–4) segments in the LVQL group from L1 to L4, and 3.5 (0–5) segments from T13 to L4 in the IQL group, showing statistically significant differences (*p* = 0.0011) (Figure 3 and Figure 6).

## 4. Discussion

The results of this study do not support our initial hypothesis, as both approaches were feasible and with a similar degree of difficulty. The studied approaches (LVQL and IQL) could promote the blockade of L1–L3 ventral branches. However, despite using high volume (0.6 mL/kg) injections, neither technique was able to consistently stain the ventral branches of the last thoracic spinal nerves (T10–T13), which are responsible for the innervation of the cranial abdominal wall [19]. Therefore, they would not be adequate to perform surgeries in the cranial abdomen. Similar findings were described in previous QL block studies carried out in dogs [2,3,4,5,6]. Additionally, the LVQL did not stain the sympathetic trunk, indicating that it could not be effective to provide visceral analgesia.

It was also hypothesized that because the site of injection selected for the LVQL approach was located closer to the skin surface, this approach could be easier to perform than the IQL. However, no differences were found comparing the quality of the images and the visualization of the needle tip of both techniques. The IQL could be more difficult to perform in larger and obese dogs, due to the increased distance from the probe to the injection site. Most of the cadavers included in our study presented a low to moderate BCS (4 out of 9). This fact, together with the wide clinical expertise of the researcher in charge of performing the ultrasound-guided administration of the injectate, could have biased these results. It must also be considered that, in one out of six hemiabdomens in the IQL approach, the tip of the needle could not be visualized. With this being a cadaveric study, the position of the tip of the needle was inferred from the feeling of the two “pop” sensations. This practice cannot be recommended in live animals, as it is mandatory to visualize the tip of the needle during the administration of the drugs to avoid vascular, neural or intracavitary injections. Other safety measures that should be undertaken before the administration of the drugs are an aspiration and a resistance to injection test, to avoid intravascular or intraneural injections, respectively [20,21].

The site of injection for the LVQL approach was selected bearing in mind its proximity with the trajectory taken by the ventral branches of the first lumbar spinal nerves. These nerves cross through the interfascial space formed by the QL and PM muscles to run along the ventral and ventro-lateral aspects of the QL, piercing the TAP [22]. Garbin et al. [3] described a lateral approach to the QL block, but their injection site was located more dorsally, in the space between the QL and the transverse process of L1.

The CT study revealed a wide distribution of injectate. In several cadavers of both groups, a caudal distribution of iopromide that reached L7 was observed. These findings are similar to previous descriptions [5,6]. In one cadaver of the IQL group, the femoral nerve was also stained. This potential complication of the QL block has been reported in humans [23], and should be considered if some degree of weakness in the hindlimbs is observed after this block. Distribution of contrast in the TAP was also observed in one hemiabdomen of the LVQL group. This spread of contrast may be the result of the new path created by the needle through the transverse abdominal aponeurosis, although an inadvertent TAP injection could not be discarded. 

The use of a larger volume of injectate than that employed by Garbin et al. [2] did not result in a further cranial dispersion of dye/contrast in any group. Descriptions from other canine cadaveric studies [2,3,4,5,6] led to similar results, even though different techniques, injection sites and volumes were evaluated. None of these studies were able to consistently reach the ventral branches of T10–T12. Thus, it could be suggested that the quadratus lumborum block would not be a suitable technique to provide analgesia of the cranial abdominal wall in dogs. For this purpose, other techniques should be considered, like the subcostal TAP block [11] or the caudal thoracic paravertebral block [24] (CTPV). Drozdzynska et al. [11] reported in a cadaveric study, that the subcostal TAP block could promote the blockade from T9 to T13. However, this technique would only be able to produce somatic analgesia [25]. Medina Serra et al. [24] described, in another cadaveric study, that the CTPV produced a great distribution of injectate between T9 and T13 also reaching the sympathetic trunk. Therefore, the latter technique could be suitable for procedures involving the cranial abdomen. 

In our study, the LVQL approach did not consistently stain the sympathetic trunk, while the IQL presented a more constant spread from L1 to L3 levels. These results show that the LVQL approach may provide a similar analgesic effect to those reported for a TAP block [9,10,12]. The LVQL could be an alternative to the TAP block, particularly in small and lean dogs, where TAP block can be difficult to perform due to the poor development of the muscles of the abdominal wall. Our results regarding the stain of the sympathetic trunk in the IQL approach, although being better than in the LVQL, are not as good as those described in previous studies that tested different approaches to perform the QL block [4,6]. Alaman et al. [4] and Marchina-Gonçalves et al. [6] described the administration of the injectate near the vertebral body of L1. This technique seems to allow a further cranial distribution throughout the sympathetic trunk, reaching T13 level in 100% [4] and 70% [6] of the cases, in comparison to the 33% observed here. The distribution of a local anesthetic up to T13 is relevant because the major splanchnic nerve emerges from the T13 ganglion and has an important role on visceral innervations [19,26]. Therefore, the presence of dye at this level, indicates that the technique could produce the blockade of this nerve, contributing to the visceral analgesia. 

In 50% of cases in the IQL group, the genitofemoral nerve was stained. This nerve is responsible for the sensitive innervation of the proximal aspect of the medial thigh, skin of pudendal region, prepuce, mammary gland, vaginal tunic and vaginal process [22,27]. A previous cadaveric study reported that the genitofemoral nerve was dyed in 40% of cases [6]. Thus, the QL block could contribute to a multimodal analgesic protocol for genitourinary surgeries, although clinical studies are necessary to confirm it.

Our study has several limitations. The results could be biased by the small number of cadavers available. The injections were carried out mostly in lean cadavers. Probably, the quality of the images obtained from larger or obese animals would be different because the ultrasonographic appearance of important structures to guide the block could be impaired in these types of patients. As the QL block is an operator-dependent technique, the fact that the ultrasound studies were performed by a researcher with high-clinical experience could also bias the results, particularly those comparing the ease of both approaches [2,3,4,5]. The freezing and thawing process could affect the tissues, as the cellular membrane could deteriorate during these processes, leading to a different distribution than what is observed in live animals. The local anesthetics could also present a distinct spread pattern compared to the mixture methylene blue/iopromide [2,3,4,5,6]. To increase the distribution of the injectate, a volume of 0.6 mL kg^−1^ per hemiabdomen was chosen. This high volume must be considered when local anesthetics are administered, in order to avoid its systemic toxicity [28,29,30,31]. So, in this blockade, the dilution of drugs would be recommended. 

## 5. Conclusions

In conclusion, the LVQL would be a suitable option to provide analgesia to the abdominal wall, but not to the abdominal viscera. Despite the high volume of injectate administered in this study, the injections did not consistently reach the ventral branches of the last thoracic nerves in any group. Therefore, none of these approaches may be recommended to provide analgesia for the cranial abdominal wall. More clinical trials are necessary to test the QL block in live animals, to determine its analgesic effects and the extension of the blockade at both somatic and visceral levels.

## Figures and Tables

**Figure 1 animals-13-02214-f001:**
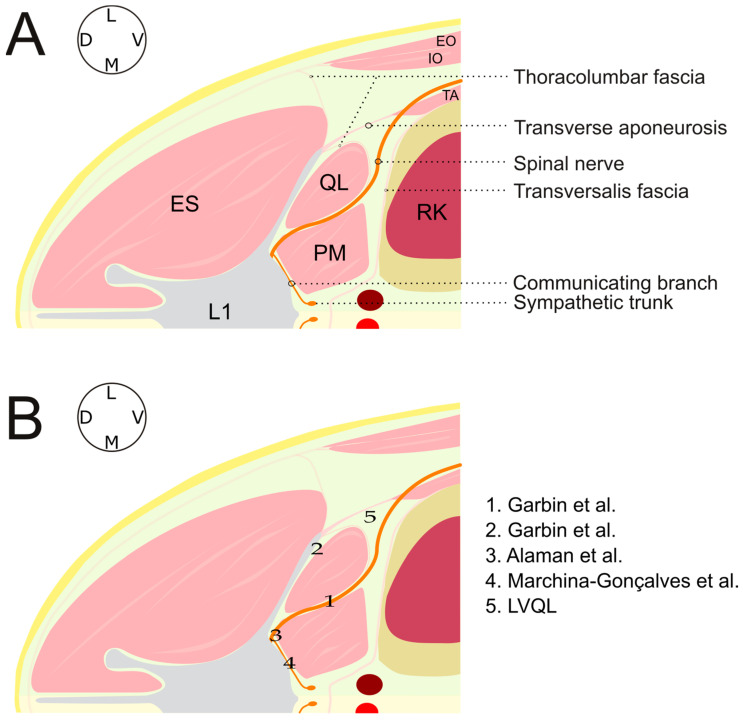
(**A**) Schematic illustration of the anatomy at the level of the first lumbar vertebra in a dog placed in left lateral recumbency. (**B**) Sites of injection of the different approaches described for a QL block in dogs [2,3,4,6]. Adapted from Marchina-Gonçalves et al. [6]. Main anatomical structures L1, first lumbar vertebra; EO, external oblique muscle; IO, internal oblique muscle; TA, transverse abdominal muscle; ES, erector spinae muscles; QL, quadratus lumborum muscle; PM, psoas minor muscle; RK, right kidney; LVQL, latero-ventral quadratus lumborum block; L, lateral; M, medial; D, dorsal; V, ventral.

**Figure 2 animals-13-02214-f002:**
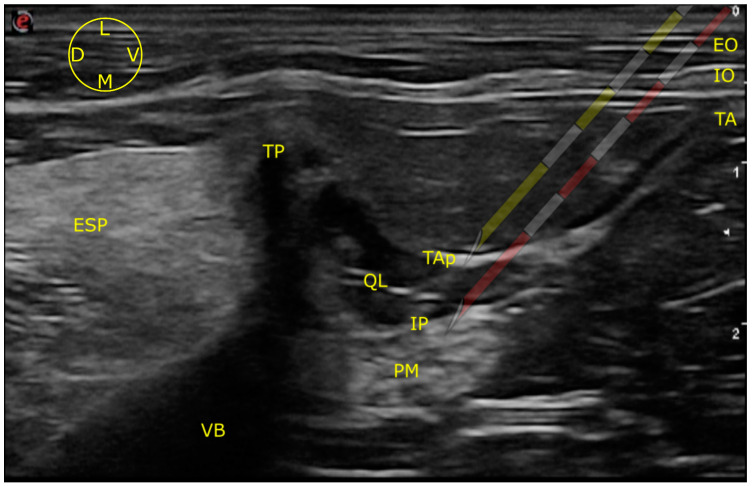
Ultrasound image of the anatomical structures identified to guide the LVQL (yellow needle) and IQL (red needle) approaches. EO external oblique muscle; ESP erector spinae muscles; IO internal oblique muscle; IP interfascial plane; PM psoas minor muscle; QL quadratus luborum muscle; TA transverse abdominal muscle; TAP transverse abdominal muscle aponeurosis; TP transverse process of L1; VB vertebral body of L1; D, dorsal; L, lateral; M, medial; V, ventral.

**Figure 3 animals-13-02214-f003:**
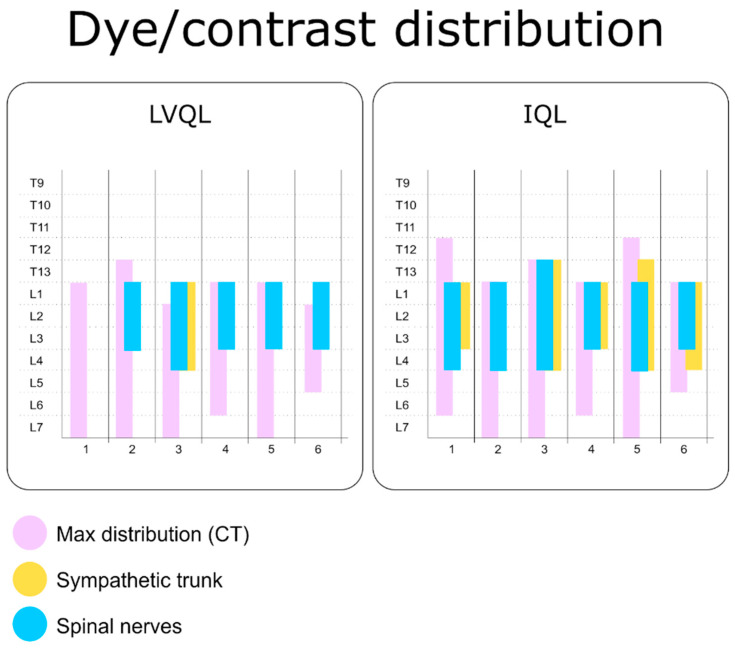
Staining of the ventral branches of the spinal nerves and the sympathetic trunk evaluated by computed tomography and anatomical dissection after the administration of 0.6 mL kg^−1^ of a mixture of methylene blue and iopromide by the LVQL and IQL approaches. LVQL, latero-ventral quadratus lumborum block; IQL, interfascial quadratus lumborum block.

**Figure 4 animals-13-02214-f004:**
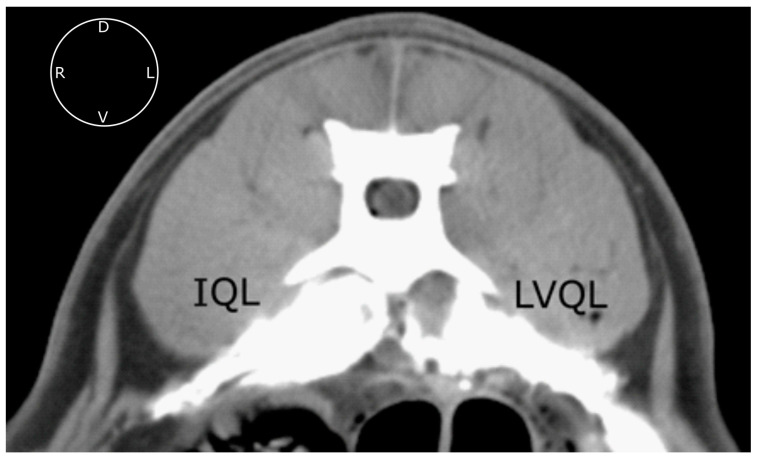
Computed tomographic image of the contrast media spread after the administration of 0.6 mL kg^−1^ of a mixture of methylene blue and iopromide by the IQL and LVQL approaches. Transverse image at L1 level with soft tissue window setting. IQL, interfascial quadratus lumborum block; LVQL, latero-ventral quadratus lumborum block; D, dorsal; L, left; R, right; V, ventral.

**Figure 5 animals-13-02214-f005:**
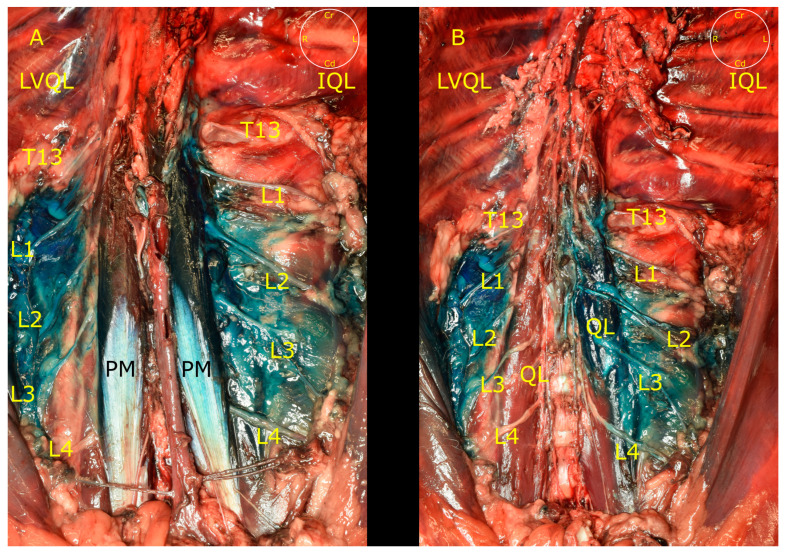
(**A**) Distribution of the dye through QL and PM muscles, and the ventral branches of the spinal nerves in the LVQL and IQL approaches. (**B**) Same dissection image after removing the psoas minor muscle (PM). IQL, interfascial quadratus lumborum block; LVQL, latero-ventral quadratus lumborum block; PM, psoas minor muscle; QL, quadratus lumborum muscle; T13, L1, L2, L3 and L4, ventral branches of T13, L1, L2, L3 and L4 spinal nerves; Cd, caudal; Cr, cranial; L, left; R, right.

**Figure 6 animals-13-02214-f006:**
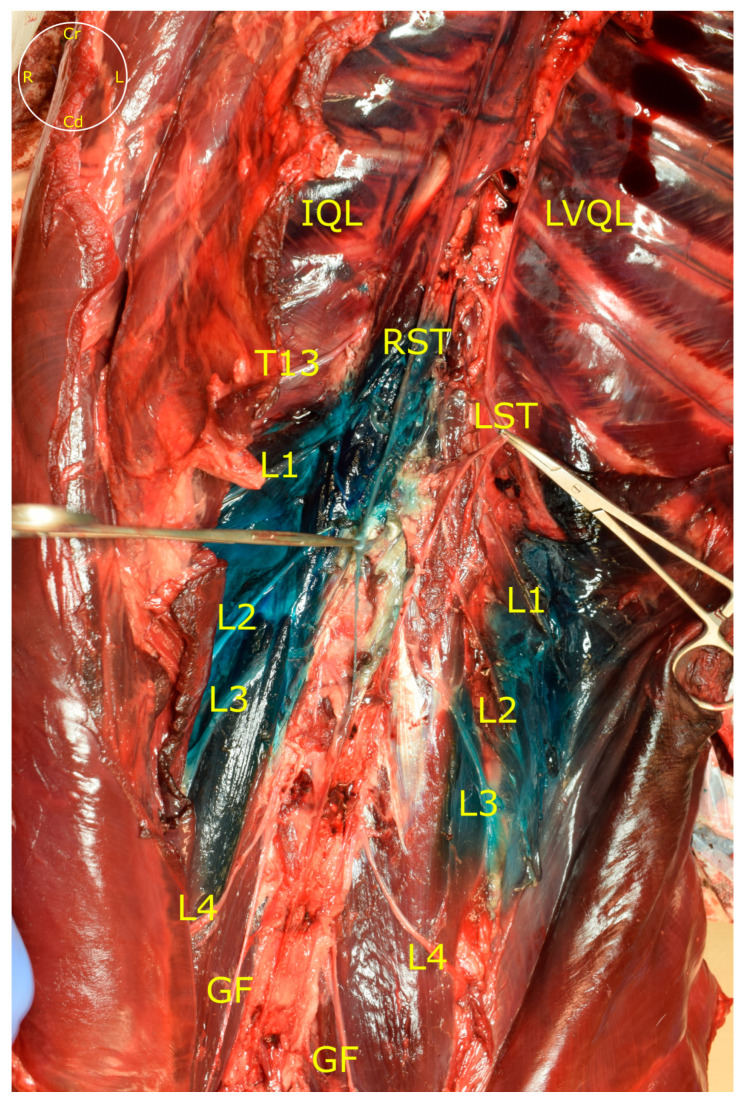
Distribution of the dye through the sympathetic truck after removing the PM and the aorta. IQL, interfascial quadratus lumborum block; LVQL, latero-ventral quadratus lumborum block; PM, psoas minor muscle; GF, genitofemoral nerve; LST, left sympathetic trunk; RST, right sympathetic trunk; T13, L1, L2, L3 and L4, ventral branches of T13, L1, L2, L3 and L4 spinal nerves; Cd, caudal; Cr, cranial; L, left; R, right.

## Data Availability

Data supporting the reported results can be sent to anyone interested by contacting the corresponding author.

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
