# Peer review of "An Ultrasound-Guided Latero-Ventral Approach to Perform the Quadratus Lumborum Block in Dog Cadavers"

_animals, 2023, doi:10.3390/ani13132214_

Round 1

Reviewer 1 Report

Dear authors,

Thank you for the opportunity to review your manuscript titled “Study of a new latero-ventral approach to perform the quadratus 2 lumborum block in dog cadavers.”. Below you will find my comments and questions about it.

Best regards!

-          Please, order alphabetically the keywords.

-          Line 18:  Instead of perforating I think it is more appropriate damaging.

-          Line 31: Replace modifications with approaches.

-          Line 40: Add ‘performing’ after ‘… the ease of’.

-    Image 1. The injection point of Alaman et al. it is not under the transverse process, it is over the vertebral body. Please modify that on the image.

-          Line 82: Replace damaging instead of perforating.

-          Lines 82-88: Add some reference, please.

-          Line 93: Please explain why the volume of 0.6 ml/kg was selected in this case.

-          Line 167: ‘Contrast’ instead of ‘contras’.

-          Lines 179 and 190: I think it is necessary to add p when it is said that the comparison is not statistically significant.

-   Line 247: How has the degree of difficulty of the technique been evaluated?

-     Line 273: Please, replace ‘concluded’ by ‘suggested’. In this work, the technique is only being evaluated in cadavers, it is not being evaluated in live animals.

-          Figure 5: Please define PM.

-          With CT, a distribution of contrast is observed up to L7 but no nerves or sympathetic trunk staining between L4-L7 (LVQL) or L3-L7 in the (IQL). Why do you think those nerves are not stained?

-   Could LVQL be an alternative to TAP rather than QL? Or maybe a modification of the TAP rather than the QL? Do you think that LVQL could have advantages over TAP?

-          Line 304-315: Regarding the limitations of the study, I think we must talk about the risk of toxicity of the anaesthetic considering the high volume used.

-          Sort abbreviations alphabetically in all the figures.

-          Review and complete the bibliographical references (3, 4, 5, 13 and 14)

Author Response

Dear Reviewer, 

Thank you for your comments.

You can find our answers in the attachment.

Kind regards

Reviewer 2 Report

Dear Authors,

thank you for reporting this data.

you will find few comments in the attachment.

kind regards
-----
the content in the attachment:

Review animals QL

Thank you, Authors, for submitting this manuscript.

Title: I don’t like to start stating “Study” it is an obviously a study, I would say “An ultrasound guided latero-ventral approach to the quadratus lumborum in dog”.

The body condition score of the cadavers is important for these techniques. <you should describe the BCS for each cadaver. Exclusion criteria of cadavers were probably skin or muscle lesion in the lateral aspect of the thoracolumbar area I suppose. You should add those information.

Another limitation is that the methylene blue and contrast medium could have a different spreading in cadavers compared to live animals. Cellular membrane could deteriorate during thawing. Furthermore, the time and temperature of thawing could also influence the quality of the structures.

You describe a single person making all the blocks but, in the limitation, you wrote that it was a team to perform the injection. How many people? Please be more precise and report who did which block and if you find out operator statistical difference in performing and reach anatomical landmarks.

You also state that in case of impossibility to see the bevel of the needle you used a double pop to detect the correct localization. I believe this is not possible and safe in live animal and I strongly suggest not to advance with the needle if you cannot see your needle. The safety of the ultrasound technique is related to the visualization of the needle while advancing so this part cannot be accepted. Please reformulate and be sure that the reader will not feel free to do that during the everyday practice. Insist on aspiration, that you never mention, and feeling of resistance of injection and visualization of the needle. Especially in this block where, as you previously mentioned, the risk of damage vital organs is high.

Author Response

(The authors gave the same response as above.)

Reviewer 3 Report

You did a nice job and wrote a pleasant paper, thank you!

Please see the attached comments.

/

Author Response

(The authors gave the same response as above.)

Round 2

Reviewer 1 Report

Dear authors,

Thank you for your responses and comments to the review. I believe that the modifications made in the second version have significantly improved the document. Congratulations for the work done!

Kind regards!

Reviewer 2 Report

Dear Authors 

Thank you for the work done on the manuscript 

I like this version much more I strongly believe that this can be a useful and interesting paper to read.

Well done